# Inflammatory, Oxidative Stress, and Apoptosis Effects in Zebrafish Larvae after Rapid Exposure to a Commercial Glyphosate Formulation

**DOI:** 10.3390/biomedicines9121784

**Published:** 2021-11-27

**Authors:** Germano Lanzarin, Carlos Venâncio, Luís M. Félix, Sandra Monteiro

**Affiliations:** 1Centre for the Research and Technology of Agro-Environment and Biological Sciences (CITAB), University of Trás-os-Montes and Alto Douro (UTAD), 5000-801 Vila Real, Portugal; cvenanci@utad.pt; 2Department of Animal Science, School of Agrarian and Veterinary Sciences, UTAD, 5000-801 Vila Real, Portugal; 3Animal and Veterinary Research Center (CECAV), UTAD, 5000-801 Vila Real, Portugal; 4Institute for Innovation, Capacity Building and Sustainability of Agri-Food Production (Inov4Agro), UTAD, 5000-801 Vila Real, Portugal; 5Instituto de Investigação e Inovação em Saúde (i3s), Laboratory Animal Science (LAS), Instituto de Biologia Molecular Celular (IBMC), University of Porto (UP), 4200-135 Porto, Portugal; 6Department of Biology and Environment (DeBA), School of Life and Environmental Sciences (ECVA), UTAD, 5000-801 Vila Real, Portugal

**Keywords:** glyphosate, Roundup, commercial formulation, inflammation, oxidative stress

## Abstract

Glyphosate-based herbicides (GBH) are the most used herbicides in the world, carrying potentially adverse consequences to the environment and non-target species due to their massive and inadequate use. This study aimed to evaluate the effects of acute exposure to a commercial formulation of glyphosate, Roundup^®^ Flex (RF), at environmentally relevant and higher concentrations in zebrafish larvae through the assessment of the inflammatory, oxidative stress and cell death response. Transgenic Tg(mpxGFP)i114 and wild-type (WT) zebrafish larvae (72 h post-fertilisation) were exposed to 1, 5, and 10 µg mL^−1^ of RF (based on the active ingredient concentration) for 4 h 30 min. A concentration of 2.5 µg mL^−1^ CuSO_4_ was used as a positive control. Copper sulphate exposure showed effectiveness in enhancing the inflammatory profile by increasing the number of neutrophils, nitric oxide (NO) levels, reactive oxygen species (ROS), and cell death. None of the RF concentrations tested showed changes in the number of neutrophils and NO. However, the concentration of 10 µg a.i. mL^−1^ was able to induce an increase in ROS levels and cell death. The activity of antioxidant enzymes (superoxide dismutase (SOD), catalase (CAT), and glutathione peroxidase (GPx)), the biotransformation activity, the levels of reduced (GSH) and oxidised (GSSG) glutathione, lipid peroxidation (LPO), lactate dehydrogenase (LDH), and acetylcholinesterase (AChE) were similar among groups. Overall, the evidence may suggest toxicological effects are dependent on the concentration of RF, although at concentrations that are not routinely detected in the environment. Additional studies are needed to better understand the underlying molecular mechanisms of this formulation.

## 1. Introduction

Glyphosate, a derivative of glycine’s (*N*-(phosphonomethyl)glycine), is the active substance in glyphosate-based herbicides (GBH). The use of GBH intensified due to the emergence of herbicide-resistant weed species [1], and it is presently the most widely used herbicide in the world [2]. Consequently, it accumulates in different areas near its application site, polluting rivers and water sources [3,4]. Glyphosate has been detected in concentrations of about 0.70 µg mL^−1^ in different regions of the world [5,6,7], with a maximum peak of 5.15 µg mL^−1^ in accidental spills [8,9]. GBHs are considered to be the pesticides with the fewest side effects available on the market, but recent studies have shown that their environmental impact may be more significant than previously thought [10], as it has shown toxic effects in different species [11,12]. However, there are still controversies about its toxicity, namely, in commercial formulations that contain adjuvants [13,14]. Some studies showed different levels of toxicity attributed to commercial formulations and their main substance in distinct aquatic species [15,16], such as the increase in oxidative stress, inhibition of immune activity [17,18], induction of lymphocyte cell dysfunction [19], neurotoxicity [15,20] and apoptosis [21]. Therefore, it is important to understand the effects of GBH formulations in a general context, especially in acute exposures, where compounds are more likely to remain undegraded [22,23].

Over the years, many new glyphosate formulations, based on different glyphosate salts and surfactants, have been introduced in the market, raising issues regarding their ecotoxicological security. Thus, it becomes pertinent to study these new formulations to assess their potential toxicity. One of the recently introduced GBHs in the European market is the Roundup^®^ Flex (RF) (MON 79351), succeeding Roundup^®^ UltraMax and maintaining the same composition as its main active principle [13]. As far as we know, no studies have been carried out with this formulation, which makes the need to assess the risks associated with the exposure to this formulation more relevant. Furthermore, there are few studies on GBHs that describe the inflammatory effect in vivo. Existing reports show the ability of these compounds to induce systemic inflammation and long-term detrimental effects in mammals and fish [24,25,26,27], demonstrating that GBH can promote changes in the innate immune response by stimulating the migration of leukocytes and also in the adaptive immune response by inducing changes in the lymphocyte response [19,27,28,29,30]. There are no reports of adverse inflammatory effects from exposure to this herbicide in zebrafish. In this context, two different strains of zebrafish larvae were selected as an animal model to clarify RF toxicity in a short exposure time. This model has shown to be a promising tool for assessing GBH toxicity [20,31,32,33,34]. One of the strains used was the transgenic Tg(mpxGFP)^i114^, which allows the visualisation of fluorescent neutrophils and infers the inflammatory response by analysing their migration and counting [35,36,37]. To avoid interactions between analyses that emit fluorescence, the wild-type AB line was used to analyse the biochemical and oxidative parameters, as well as the evaluation of cell death.

## 2. Materials and Methods

### 2.1. Chemicals and Solutions

The commercial formulation, Roundup^®^ Flex (RF; Bayer CropScience, Carnaxide, Portugal), of which the active ingredient (a.i.) is glyphosate potassium (CAS 70901-12-1) with 35.5 wt% by weight, was used in this study. This formulation also contains the adjuvant ether amine Ethoxylate 6% (CAS 68478-96-6). The concentrations used, which were 1, 5 and 10 µg a.i. mL^−1^ (5, 25 and 50 µM, respectively), were prepared from a stock solution of RF 100 µg a.i. mL^−1^ (500 µM), taking into consideration the glyphosate concentration present. It is important to note that, as far as we know, the highest concentration used is twice the environmentally relevant concentration. Dilutions were made with E3 medium (0.5 mM NaCl, 0.017 mM KCl, 0.033 mM CaCl_2_, 0.033 mM MgSO_4_, pH 7.0–7.4). Copper, added as copper sulphate pentahydrate (CuSO_4_.5H_2_O, CAS 7758-99-8 from Merck, S.A, Lisbon, Portugal), was prepared from a stock solution of 1000 µg mL^−1^ (4 mM) and further diluted in E3 medium to 2.5 µg mL^−1^ (10 µM), according to a previous study [38]. Tricaine methanesulfonate (MS-222) (CAS 886-86-2 from Sigma Aldrich, Lisbon, Portugal), used as an anaesthetic with a stock solution of 1500 mg L^−1^ (5.74 mM), was prepared in distilled water and buffered to pH to 7.4 with sodium bicarbonate, and further diluted in E3 medium to 150 mg L^−1^ (574 µM) according to [39,40,41]. All other reagents used were purchased from Sigma (Lisbon, Portugal).

### 2.2. Maintenance and Reproduction of Zebrafish

The Tg(mpxGFP)^i114^ and wild-type AB adult zebrafish (*Danio rerio*) were maintained in the fish facilities of the University of Trás-os-Montes and Alto Douro (Vila Real) in glass aquariums with tap water from the city of Vila Real, which was dechlorinated, aerated, charcoal-filtered and UV-sterilised at 28 °C (pH 7.5–8). The trasngenic line was obtained from the European Zebrafish Resource Center (EZRC, Karlsruhe Institute of Technology, Eggenstein-Leopoldshafen, Germany). The maintenance and reproduction of the zebrafish were carried out in accordance with the Portuguese legislation (DL 133/2013 from 7 August 2013) and the European Directive 2010/63/EU on animal welfare (2010/63/EU from 22 September 2010). Adult fish were fed twice a day with a commercial diet (Zebrafeed, Sparos, Olhão, Portugal), and the facilities were maintained in a circadian cycle (14 h light/10 h dark). The reproductive activity was promoted by the union of zebrafish in a proportion of 1 female: 2 males, the spawning was induced by the morning light and the eggs were collected, bleached and washed with chloramine-T (0.5% *w*/*v*) dissolved in E3 medium [42,43]. The embryos were observed in a SMZ 445 stereomicroscope (Nikon, Tokyo, Japan), and those fertilised and with normal morphology were randomly selected and kept in 6-well plates with E3 medium in a water bath at 28.5 °C until 72 h post-fertilisation (hpf).

### 2.3. Tail Transection, Neutrophil Migration Count

To evaluate the neutrophil migration response, the analyses were carried out using 72 hpf Tg(mpxGFP)^i114^ larvae, following the previously described methods [44,45,46] with some modifications. The treatment and evaluation scheme are described in Figure 1A. Briefly, 30 min before the tail transection, the larvae were exposed to RF concentrations (1, 5 and 10 µg a.i. mL^−1^). As a positive control, and as already described in zebrafish larvae as a pro-inflammatory agent [47], a concentration of CuSO_4_ 2.5 µg mL^−1^ was used. The control group was maintained in E3 medium. At least five replicates of five larvae were used per group. Subsequently, the larvae were anesthetised by immersion in MS-222 (150 mg L^−1^), and the tail was transacted with a sterile scalpel. Posteriorly, the larvae were washed in E3, placed in their respective solutions and exposed for another 4 hpi (hours post injury). Afterwards, the larvae were transported to the microscopy laboratory for image capture (Figure 1B). The fluorescent images were obtained in an inverted microscope (IX 51, Olympus, Antwerp, Belgium), using a 4X Olympus UIS-2 objective lens (Olympus Co., Ltd., Tokyo, Japan) equipped with an Olympus U-RFL-T fluorescent light source (Olympus, Antwerp, Belgium) and FITC filter. Data were acquired through the Cell A software (Olympus, Antwerp, Belgium), and the images were processed with Adobe Photoshop CS6 (Adobe Systems, San Jose, CA, USA). The quantification of the number of neutrophils migrated to the cut site (area of 250 µm from the cut was selected) was performed using an automatic cell count extension (Find Maxima) from the ImageJ2 program (version 2.0.0, National Institutes of Health of the USA, Bethesda, USA) [48].

### 2.4. Biochemical Parameters in WT Larvae

To evaluate the biochemical parameters, 72 hpf WT larvae were exposed to RF (5 and 10 µg a.i. mL^−1^) for 4 h 30 min. As a positive control, the pro-inflammatory agent used was CuSO_4_ (2.5 µg mL^−1^). The control group was maintained in E3 medium. At least five replicates of 50 larvae were performed. The treatments and evaluation scheme are described in Figure 2A. After the exposure, the larvae were washed and frozen at −80 °C in 0.32 mM de sucrose, 20 mM de HEPES, 1 mM de MgCl_2_ and 0.5 mM of phenylmethyl sulfonylfluoride (PMSF), pH = 7.4. The samples were then processed as previously reported [34,49] by homogenising them using the TissueLyzer II apparatus (Qiagen), centrifuging for 20 min at 15,000× *g* (12,517 rpm) at 4 °C, and collecting the supernatants for further analysis in duplicate using a PowerWave XS2 microplate scanning spectrophotometer (Bio-Tek Instruments, Vermont, WI, USA) or Varian Cary Eclipse (Varian, Santa Clara, CA, USA) spectrofluorometer equipped with a microplate reader. To determine the reactive oxygen species (ROS) levels, the probe 2′,7′-dihlorofluoresceindiacetate (DCFH-DA) was used as previously described [50], with readings at 485 nm (excitation) and 530 nm (emission). The ROS levels were expressed in nmol of DCF mg protein^−1^, based on a standard DCF curve (0–100 µM). The activity of the superoxide dismutase enzyme (SOD) was verified by the inhibition of nitrobluetetrazolium at 560 nm, as described elsewhere [51], and based on a SOD standard curve (0–5 U mL^−1^). The results are expressed in U mg protein^−1^. To determine the catalase activity (CAT), the methodology was as according to Aebi 1984 [33] at 240 nm based on a standard bovine catalase curve (0–3 U mL^−1^). The results are presented as U mg protein^−1^. Glutathione peroxidase (GPx) was analysed at 340 nm [52], using an extinction coefficient of 6.22 mM^−1^ cm^−1^, and the results were presented in µmol NADPH min mg protein^−1^. Through fluorometric measurements, the levels of reduced glutathione (GSH) and oxidised glutathione (GSSG) were determined by derivatisation with ortho-phthalaldehyde [53] at 320 nm excitation wavelength and 420 nm emission. The results were estimated with a standard curve GSH and GSSG (0–100 µM). The levels are expressed in µmol GSH or GSSG per mg of protein. The oxidative stress index (OSI) was calculated according to the ratio between GSH and GSSG. Glutathione S-Transferase (GST) activity was analysed at 340 nm [54] by the conjugation reaction between 2,4-dinitrochlorobenzene (CDNB, extinction coefficient of 9.6 mM^−1^ cm^−1^) and GSH. The results are expressed in µmol min mg protein^−1^. For the measurement of lipid peroxidation intensity (LPO), the methodology described before [55] was used by reading the thiobarbituric acid reactive substances (TBARS) at 530 nm and 600 nm (non-specific) and using a standard curve (0–100 µM) of malondialdehyde bis (dimethyl acetal). The results are expressed in mmol MDA mg protein^−1^. Lactate dehydrogenase (LDH) activity was determined at 340 nm [56], using the extinction coefficient of 6.22 mM^−1^ cm^−1^. The results are expressed in nmol NADH min mg protein^−1^. Acetylcholinesterase (AChE) activity was determined at 405 nm [57] using the extinction coefficient of the 5-thio-2-nitrobenzoic acid of 13.6 mM^−1^ cm^−1^. The results are expressed in nmol TNB min mg protein^−1^. The level of nitric oxide (NO) was determined using the Griess method, as described previously [58,59], with some modifications. Briefly, the samples were mixed with the Griess reagent in a 1:1 ratio and incubated for 15 min at room temperature, after which the absorbance was read at 546 nm. Sodium nitrate was used to construct a standard curve (0–1 µM). The results are reported in nmol NO mg protein^−1^. To normalize data, protein quantification was performed at 280 nm using the Take3 Multi-Volume plate (Take3 plate, BioTek Instruments, Vermont, WI, USA).

### 2.5. Intracellular Analysis of Apoptosis in WT Larvae

The analysis of cell death was performed in 72 hpf WT larvae exposed to RF (5 and 10 µg a.i. mL^−1^) and CuSO_4_ (2.5 µg mL^−1^) separately for 4 h 30 min by incubation in an acridine orange (AO) probe, as described before [19,60]. The control group was maintained in E3 medium. The exposures and evaluation scheme are described in Figure 3A. At least five replicates of 25 larvae were incubated in the dark in an AO solution (5 µg L^−1^) for 30 min at 28 °C, with subsequent washes being performed before sampling. The larvae were homogenised with the buffer used for biochemical analysis, followed by homogenisation and centrifugation as described above, after which the supernatant was reused and transferred to a specific reading plate. Readings were taken through the Varian Cary Eclipse spectrofluorometer (Varian, Santa Clara, CA, USA), and the fluorescence intensity was measured at the excitation and emission wavelengths of 488/515 nm, respectively. The levels of induced apoptosis are expressed as a percentage of the control. Before this process, some larvae were separated and used to capture illustrative fluorescent images (Figure 3B) in an inverted microscope (IX 51, Olympus, Lisboa, Portugal) equipped with an Olympus U-RFL-T fluorescent light source and FITC filter, using a 4X Olympus UIS-2 objective lens (Olympus, Lisboa, Portugal). Data were then acquired using Cell^A software (Olympus, Lisboa, Portugal).

### 2.6. Statistics

Statistical tests were performed using the IBM SPSS statistics program 26 for Windows (SPSS Inc., Chicago, IL, USA). The statistical tests were applied according to the normality (Shapiro–Wilk test) and homogeneity of variance (Levene’s test) tests. When the data followed the normal distribution, the differences between groups were assessed by a unilateral analysis of variance (ANOVA) followed by Tukey’s multiple comparison test and data expressed as mean ± standard deviation. When these assumptions were not fulfilled, data analysis was performed using non-parametric Kruskal–Wallis analysis of variance, followed by the Dunn test with a Bonferroni correction for multiple comparisons, and data expressed as medians and interquartile range (25th and 75th percentiles). The differences were defined with *p* < 0.05.

## 3. Results

### 3.1. RF Does Not Induce Neutrophil Migration in Tg(mpxGFP)^i114^ Larvae

To assess the possible effects on inflammation induction, the number of neutrophils that migrated to the injury site after 4 h of the tail fin cut and after exposure to 1, 5 and 10 µg a.i. mL^−1^ of RF were analysed, and the results are presented in Figure 1B,C. Copper sulphate exposure induced a 75% increase in the number of neutrophils that migrated to the tail relative to the control (*p* = 0.0001). Copper also showed an increase in relation to RF concentrations, 1, 5 and 10 µg a.i. mL^−1^ (*p* = 0.0001). RF exposures, in all concentrations tested, were not able to induce an increase in the number of neutrophils compared to control (F(4, 22) = 19.80, *p* < 0.0001).

### 3.2. Higher RF Concentration Induces an Increase in ROS Levels in WT Larvae

The results of the analysis of biochemical parameters to assess oxidative stress caused in 72 hpf zebrafish WT larvae, exposed to RF concentrations and copper sulphate, separately, for 4 h 30 min, that showed statistical differences, are shown in Figure 2B. The remaining analyses are shown in Appendix A. ROS levels (Figure 2B) increased significantly in the copper sulphate-exposed larvae when compared with the control (*p* < 0.0001) and with larvae exposed to the RF concentrations (µg a.i. mL^−1^) of 1 (*p* = 0.007) and 5 (*p* = 0.004). Exposure to the highest concentration of RF (10 µg a.i. mL^−1^) caused a significant increase in ROS levels compared to the control group (*p* = 0.004). Regarding SOD activity (Figure 2B), larvae exposed to RF concentrations of 5 µg a.i. mL^−1^ presented higher values than the copper sulphate-exposed (*p* = 0.03) and RF at 1 µg a.i. mL^−1^ (*p* = 0.01). No differences were observed between the RF concentrations and the control group. In CAT and GPx, no changes were observed in both groups. Considering glutathione, the GSH levels (Figure 2B) of larvae exposed to RF 5 µg a.i. mL^−1^ were significantly lower than the ones observed in the copper sulphate-exposed (*p* = 0.01) and the RF 1 µg a.i. mL^−1^ (*p* = 0.04) groups. On the other hand, for GSSG (Figure 2B), no changes were detected among groups. The OSI values (Figure 2B) of larvae exposed to RF 10 µg a.i. mL^−1^ were significantly inferior to those of copper sulphate-exposed (*p* = 0.02) and RF 1 µg a.i. mL^−1^ (*p* = 0.04) exposed animals. Regarding the levels of LPO (Figure 2B), the concentration of RF 5 µg a.i. mL^−1^ showed significant changes, with increased values when compared to copper sulphate-exposed larvae (*p* = 0.04). There were no differences between groups in the activity of the enzymes GST, LDH and AChE. The NO levels (Figure 2B) were higher in the copper sulphate-exposed larvae when compared to the control (*p* = 0.01) and RF 1 µg a.i. mL^−1^ (*p* = 0.03) groups.

### 3.3. Higher RF Concentration Induces Apoptosis In Vivo

As shown in Figure 3C, the distribution of apoptotic cells after copper sulphate exposure (*p* = 0.003) and in the highest concentration of RF (10 µg a.i. mL^−1^) (*p* = 0.009) showed increased levels compared to the control (X^2^(4,20) = 11.269, *p* = 0.024).

## 4. Discussion

GBH concentrations continue to be found in the environment, directly influenced by factors such as their excessive use and inadequate management in agricultural and industrial practices [61]. This has led to a recently documented increase in glyphosate residues in different ecosystems being detected in soil, water, foods and humans [4,62,63,64]. This study evaluated the changes induced by rapid exposure to a formulation of GBH (Roundup^®^ Flex) regarding neutrophil migration, biochemical parameters of oxidative stress and early cell death, using zebrafish larvae as an animal model. As far as we know, this is the first study to evaluate this new glyphosate formulation and its effects on neutrophil migration in zebrafish. The results of the acute exposure showed that, under these conditions, the tested concentrations did not induce changes in the migration of neutrophils. On the other hand, the highest concentration tested (10 µg a.i. mL^−1^) increased the ROS levels and early cell death. The exposure with copper sulphate (proinflammatory control) showed an increase in the number of neutrophils migrating to the lesion and increased levels of ROS, NO and cell death.

As is already known, the innate immune system in the early life of zebrafish is mainly composed of neutrophils and macrophages, which play a notorious defensive role. Neutrophils are the first to be recruited to the injury site [65]. In zebrafish, neutrophil migration increases considerably about 3 h after the injury [66], peaking at 4 h [67]. The Tg(mpxGFP)^i114^ larvae are effective for testing inflammation mechanisms [68,69] due to the fluorescence properties of their neutrophils. Monitoring the migration of neutrophils to a damaged area can be useful to quickly assess the inflammatory response and identify pollutants capable of inducing inflammation [70]. Indeed, the exposure to copper sulphate (proinflammatory control) generated an increase in the number of neutrophils migrating to the injury site, as expected [35,47]. In this study, and compared to the control group, none of the concentrations of RF used were able to induce an increase in neutrophil migration to the injured site. In contrast, other studies have shown the induction of pro-inflammatory states with exposure to glyphosate-containing compounds [71,72].

The assessment of the environmental safety of a pesticide is essential to ensure the protection of ecosystem life [35]. Several studies have shown that pollutants trigger oxidative stress and apoptosis in different fish species [73,74,75,76]. Oxidative stress occurs due to an imbalance between ROS production and the availability of antioxidants or radical scavengers [77]. This imbalance promotes oxidation that results in structural modification of biomolecules, such as carbohydrates, lipids, proteins and nucleic acids, consequently causing cellular alterations that can lead to high levels of inflammation and even cell death [78,79,80]. GBHs confer cytotoxic and genotoxic effects, increase oxidative stress, alter the immune system, impair some brain functions and are, supposedly, correlated with the development of some types of cancer [71,72,81,82,83,84]. In our study, the positive control (copper sulphate) behaved as expected [85,86], increasing the levels of ROS, NO and cell death in WT larvae. Concerning RF exposures, the concentration of 10 µg a.i. mL^−1^ caused an increase in ROS levels, although no other changes occurred in the remaining oxidative stress parameters analysed. This was probably due to the short exposure time, since recent studies, with longer exposure time to GBH and glyphosate at similar concentrations, and in different species, reported alterations of antioxidant enzymes’ action, secondary to high levels of ROS [32,87,88,89]. High levels of ROS are also associated with the induction of cell death [90,91] through the cellular damage caused by free radicals and the triggering of pro-apoptotic signals [92]. The apoptotic effects of RF were assessed through AO staining. This is a selective metachromatic dye of nucleic acid that emits green fluorescence after intercalation with DNA. The rationale is justified by the fact that, in apoptotic cells, acridine orange may permeate them and bind to DNA, whereas normal cells are not permeable to it [93,94]. This technique has been widely used to detect apoptosis in zebrafish exposed to different types of compounds [49,95]. The data found by AO staining showed that the highest concentration of RF could exert pro-apoptotic activity, the same concentration group that showed high levels of ROS. Similar results were described in a study performed with glyphosate during the embryonic development of zebrafish, showing a relation between the high levels of ROS and apoptosis induction [11]. These findings probably occurred due to the RF cytotoxic effect, where the increased production of ROS activates apoptotic pathways unable to alter cellular antioxidant processes [91,92], since no changes were verified in the other analyses related to oxidative stress. In a direct relationship to human cell lines, glyphosate demonstrated the ability to induce a significant increase in MDA levels, NO production, ROS and caspase 3/7 activity, altering the expression of 24 genes related to apoptosis [96].

## 5. Conclusions

The results obtained in the present study show that a rapid exposure to RF at a concentration of 10 µg a.i. mL^−1^ increased the production of ROS in WT larva and triggered cellular apoptosis. Although this concentration is higher than those considered environmentally relevant, acute exposure may have limited the potential toxic effect of the lower concentrations tested. Therefore, at environmentally relevant concentrations, RF did not induce toxicity in zebrafish larvae in the parameters analysed. Changes in ROS formation and apoptosis induced by GBH exposure can have important pathological consequences for living organisms. Thus, a better clarification of the potential toxic mechanism is still needed to optimize the regulation of the use of GBH worldwide.

## Figures and Tables

**Figure 1 biomedicines-09-01784-f001:**
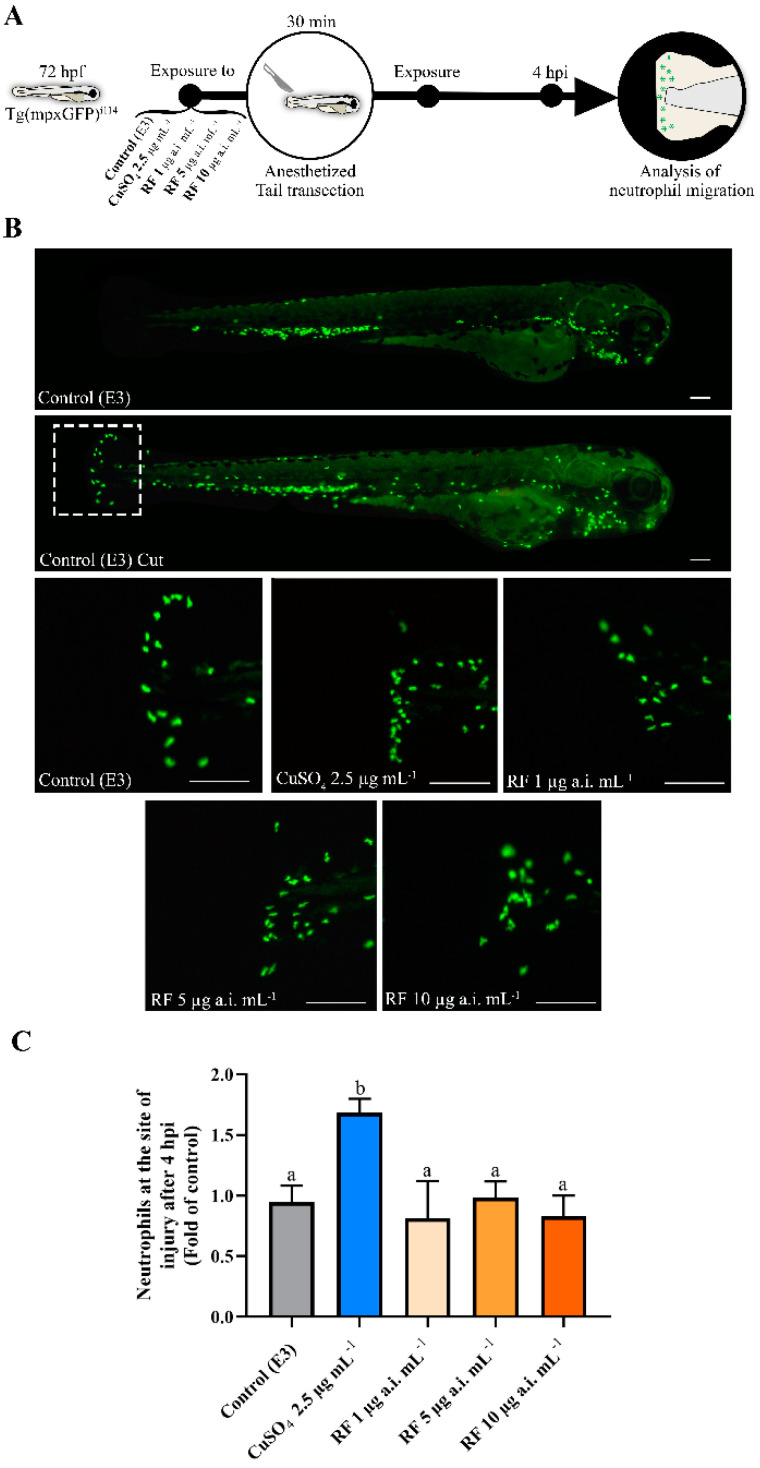
Treatments’ effect on neutrophil recruitment to the lesion site in zebrafish (Tg(mpxGFP)^i114^) larvae induced by tail transection. Data from at least five independent samples of five random animals each. (**A**) Illustration of the experimental protocol of the neutrophil migration study. (**B**) Illustrative image of a normal Tg(mpxGFP)i114 (control) and a larva with transection of the tail (control cut). Detailed images of the local site of the transection after exposure to treatments during 4 hpi. The scale bar represents 125 μm. (**C**) Number of neutrophils migrated to the tail at 4 hpi. Values were normalised according to the control group. Data are expressed as mean ± SD and statistical analysis was performed using a one-way ANOVA followed by Tukey’s multiple comparison test. Different letters represent statistical differences among treatment groups (*p* < 0.05).

**Figure 2 biomedicines-09-01784-f002:**
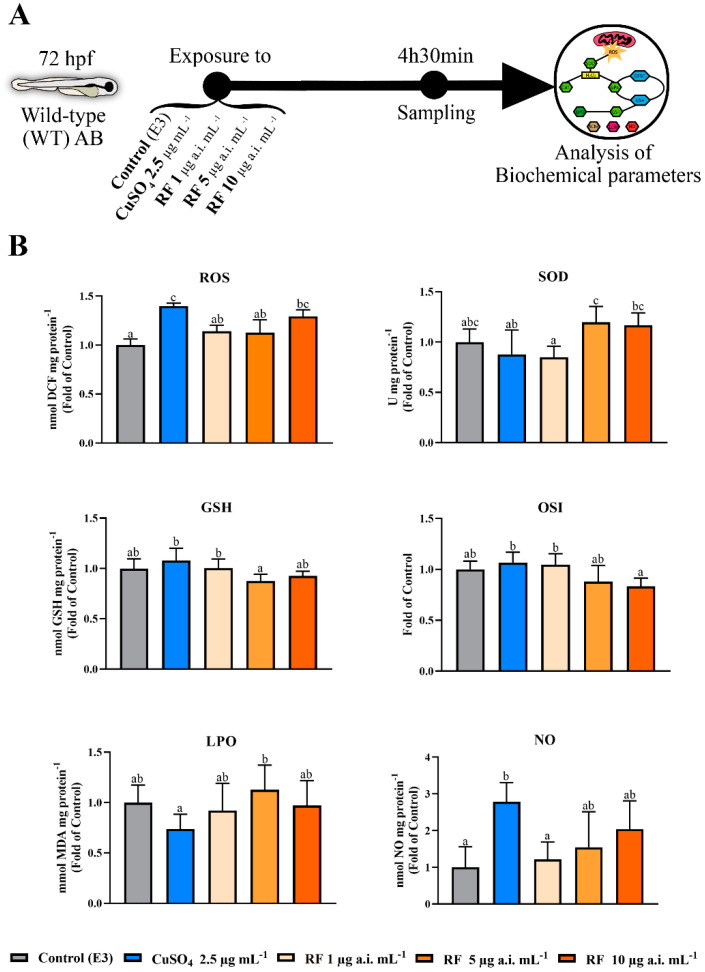
Effect of the different chemicals in biochemical parameters evaluated after exposure for 4 h 30 min in 72 hpf zebrafish WT larvae. Data from at least five independent samples of 50 random animals each. (**A**) Schematic diagram showing the study experimental protocol of biochemical parameters. (**B**) Graphs of biochemical parameters that showed significant differences after exposure to different chemicals. Values were normalised to the control group. Data are expressed as mean ± SD for parametric data distribution or median (interquartile range) for non-parametric data. Statistical analysis was performed using one-way ANOVA followed by Tukey’s multiple-comparison test or Kruskal–Wallis followed by Dunn’s test. Different letters represent statistical differences among treatment groups (*p* < 0.05).

**Figure 3 biomedicines-09-01784-f003:**
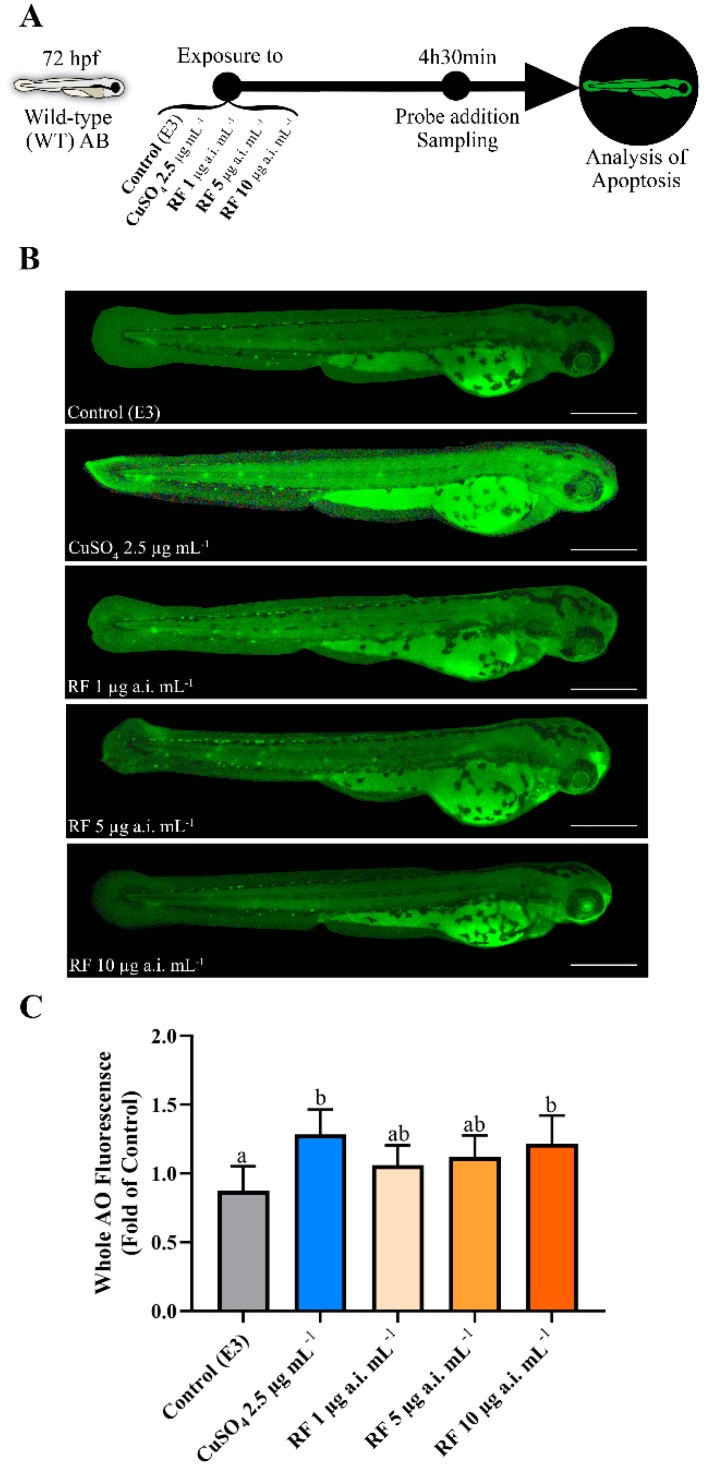
Effect of different chemicals in cell death induction, evaluated after exposure for 4 h 30 min in 72 hpf zebrafish WT larva. Data from at least five independent samples of 25 random animals each. (**A**) Schematic diagram showing the study’s experimental protocol to cell death evaluation. (**B**) Illustrative images from larvae exposed to the AO probe. Scale bar represents 500 μm. (**C**) Result of AO fluorescence intensities in homogenised larvae. Values were normalised according to the control group. Data are expressed as mean ± SD and statistical analysis was performed using a one-way ANOVA followed by Tukey’s multiple comparison test. Different letters represent statistical differences among treatment groups (*p* < 0.05).

## Data Availability

Not applicable.

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
