# Peer review of "Inflammatory, Oxidative Stress, and Apoptosis Effects in Zebrafish Larvae after Rapid Exposure to a Commercial Glyphosate Formulation"

_biomedicines, 2021, doi:10.3390/biomedicines9121784_

Round 1

Reviewer 1 Report

Specific comments

1) Title: ... and apoptosis of zebrafish, what do you mean? you may mean apoptosis occurred in zebrafish larvae.

2) Abstract:(WT) zebrafish larvae (72 hours post-fertilization) were exposed to 1, 5, and 10 μg mL-1 of RF (based on the active ingredient concentra tion) for 4:30 h. What meaning is 4:30 h?The authors should explain it in the text.

3) The name of journal should be capitalized the first letter, for example, 16. Fish physiology and biochemistry, 19. Fish & shellfish immunology, etc.

4) The authors should check the English grammar and appropriately improve the writing.

Author Response

1) Title: ... and apoptosis of zebrafish, what do you mean? you may mean apoptosis occurred in zebrafish larvae.

Response: In order to clarify the idea, the tile was changed.

2) Abstract:(WT) zebrafish larvae (72 hours post-fertilization) were exposed to 1, 5, and 10 μg mL-1 of RF (based on the active ingredient concentration) for 4:30 h. What meaning is 4:30 h?The authors should explain it in the text.

Response: According to the reviewer concern, the text was elucidated.

3) The name of journal should be capitalized the first letter, for example, 16. Fish physiology and biochemistry, 19. Fish & shellfish immunology, etc.

Response: The reference style was revised according to the guide for authors.

4) The authors should check the English grammar and appropriately improve the writing.

Response: The text was revised by a native English person in order to correct errors and typos.

Reviewer 2 Report

In the manuscript Biomedicine-1468040, by Lanzarin et al, authors have examined the toxicity readouts in zebra fish model against acute exposure to environmentally relevant and higher concentrations of a newly introduced glyphosate formulation “Roundup ® Flex/RF”. They examined inflammatory response using neutrophils migration as a readout. They further examined oxidative stress readouts such SOD, MDA, LPO, ROS and cell death in exposed Zebra fish. For comparison, they used a Copper sulphate as positive control. None of the tested conc. of RF exerted inflammatory response. Only higher conc induced oxidative stress and cell death in the exposed animals. Overall, study is a short presentation of some inflammatory response and oxidative stress readouts but lacks in terms of advancement of the knowledge. With more mechanistic results/finding it can be a significant study.  

I have following concerns-

  1. Although the study is highlighted as the first study to show the toxicity potential of the new formulation, it still shows the similar parameters. In general, it would be obvious to have these toxicity parameters as the principle active in all formulations is the same. Furthermore, inflammatory readout (number of migrated neutrophils) was not affected in any of the tested concentrations and the toxic effects were observed only at very high concentration. The presented glyphosates induced toxicity readouts have already been reported in previous reports. No further work to understand the mechanism is there in this study. So, this makes study superficial and lacks advancement to the prior knowledge.

  1. Title- Since no effect on neutrophil migration was observed against tested glyphosate formulation concentrations, please rephrase the title as it misleads.

  1. How does the newly introduced formulation “Roundup® Flex (MON79351) is different from the other GBH? As per the description, it seems all the GBH formulations have same principle active compound (glyphosate). This makes it obvious to similar toxicity effects reported in previous reports.  

  1. Methods sections (biochemical parameters) are not well described.

  1. Heading of the Result sections are not informative. Please rephrase. For example, “Neutrophils in Tg(mpxGFP)i114 larvae”. What does this result heading convey?

  1. Presentation of statistical comparison using letters is not easy to follow up. Especially using single, double, and triple letter code. Please simply this. Asterisks can be used.

  1. Please use italic “p” for p- value as per the statistical standard throughout the manuscript.   

  1. Result section 3.3. Please rephrase “Apoptosis generation”.

  1. In the discussion section, first sentence says “GBH concentrations found in the environment are increasing…”. It is widely used because of its degradability. So, it might not be accurate to say if its concentration is increasing. Unless there are reports for its environmental accumulation.

  1. Please explain “the different ecosystems” in the line 264-265.

  1. Line 265, “In this study, were evaluated”. Please check for correctness of the sentence.

  1. Line 268-269. “As far we know, this is the first study to evaluate this glyphosate formulation on migrating neutrophils in zebrafish.” What does this sentence imply? Is it the first study to use neutrophil migration readout? or is it the first study evaluating neutrophil migration against glyphosate exposure? Based on using this readout against specifically this new formulation is not enough to highlight it as first study. Especially all formulations have the same active principal component. Please rephrase or consider exclude the sentence accordingly or provide explanation.

  1. Please replace “Copper exposure” to “copper sulphate exposure” throughout the manuscript. Copper sulphate is a positive control for oxidative stress.

  1. Conclusion section is repetitive of result. It mostly talks about future studies needed. Having the answers for the raised issues would have been an advancement in the previous knowledge.
  2. Please rephrase the sentence “it will allow for a ……” in line 329-330.

  1. Please revise manuscript thoroughly for typos and syntax errors.

Author Response

1) Although the study is highlighted as the first study to show the toxicity potential of the new formulation, it still shows the similar parameters. In general, it would be obvious to have these toxicity parameters as the principle active in all formulations is the same. Furthermore, inflammatory readout (number of migrated neutrophils) was not affected in any of the tested concentrations and the toxic effects were observed only at very high concentration. The presented glyphosates induced toxicity readouts have already been reported in previous reports. No further work to understand the mechanism is there in this study. So, this makes study superficial and lacks advancement to the prior knowledge.

Response: We thank the reviewer for the suggestion and recognize that a broad assessment may be useful for the mechanistic understanding of exposure to glyphosate and its formulations. The glyphosate substance has been on the market for a long time and, being the most used herbicide in the world, it is widely studied. However, over the years, many new glyphosate formulations have been introduced based on different glyphosate salts and surfactants raising issues regarding their ecotoxicological security. Thus, it becomes pertinent to continue to study these new formulations, even with similar experimental approaches. This information was added in the text for clarification, highlighting the relevance of the study. However, and not being the objective of this work, the use of further evaluations would require laboratory resources that, unfortunately, are not possible to be carried out at the moment. Thus, the authors believe that the study hypothesis strengthens the purpose of the research carried out and kindly ask for the manuscript to be accepted in its current format.

2) Title- Since no effect on neutrophil migration was observed against tested glyphosate formulation concentrations, please rephrase the title as it misleads.

Response: As suggested by both reviewers, the title was changed.

3) How does the newly introduced formulation “Roundup® Flex (MON79351) is different from the other GBH? As per the description, it seems all the GBH formulations have same principle active compound (glyphosate). This makes it obvious to similar toxicity effects reported in previous reports.  

Response: Thanks for the comment. In fact, glyphosate is the main active substance in GBH, however Roundup formulations marketed by the company Monsanto (Bayer) have periodically changed to include surfactants with a favorable ecotoxicological profile. Yet, different Roundup formulations result in different fish toxicological profiles (doi: 10.1016/j.fct.2019.03.053) requiring the continues study of the new formulations released. Yet, until now, this formulation was not studied in non-target species which led us to develop this study.

4) Methods sections (biochemical parameters) are not well described.

Response: In order to avoid plagiarism, the methods were only briefly described with the inclusion of two references from the group where these methods are detailed described. We hope that the reviewer accept the manuscript in the current format.

5) Heading of the Result sections are not informative. Please rephrase. For example, “Neutrophils in Tg(mpxGFP)i114 larvae”. What does this result heading convey?

Response: As suggested by the Reviewer, the titles in the results sections have been revised.

6) Presentation of statistical comparison using letters is not easy to follow up. Especially using single, double, and triple letter code. Please simply this. Asterisks can be used.

Response: The authors appreciate the comment raised by the Reviewer. However, the authors have to disagree with the reviewers as, for multiple comparisons, as is the case in the present study, the use of different letters allow efficient reporting of statistical differences (doi: 10.2134/agronj2017.10.0580,10.1161/CIRCULATIONAHA.107.700971, 10.1016/j.csda.2006.09.035, among others) with different letters (a, b, c, d) standing for significant differences (p<0.05). Therefore, the authors believe the statistical presentation used is correct and the better way to evidence the statistical differences.

7) Please use italic “p” for p- value as per the statistical standard throughout the manuscript.

Response: Thanks for the comment, as suggested by the Reviewer, we have revised the text and changed the “p” in p- values throughout the manuscript to italics.

 8) Result section 3.3. Please rephrase “Apoptosis generation”.

Response: As suggested by the Reviewer, the results headings were revised.

9) In the discussion section, first sentence says “GBH concentrations found in the environment are increasing…”. It is widely used because of its degradability. So, it might not be accurate to say if its concentration is increasing. Unless there are reports for its environmental accumulation.

Response: As suggested by the Reviewer, the sentence “GBH concentrations found in the environment are increasing…”, was changed to “GBH concentrations continue to be found in the environment…”.

 10) Please explain “the different ecosystems” in the line 264-265.

Response: As suggested by the Reviewer, the text was revised.

 11) Line 265, “In this study, were evaluated”. Please check for correctness of the sentence.

Response: As suggested by the Reviewer, the sentence has been removed and rephrased.

12) Line 268-269. “As far we know, this is the first study to evaluate this glyphosate formulation on migrating neutrophils in zebrafish.” What does this sentence imply? Is it the first study to use neutrophil migration readout? or is it the first study evaluating neutrophil migration against glyphosate exposure? Based on using this readout against specifically this new formulation is not enough to highlight it as first study. Especially all formulations have the same active principal component. Please rephrase or consider exclude the sentence accordingly or provide explanation.

Response: As suggested by the reviewer, we reformulated the phrase “As far as we know, this is the first study to evaluate this glyphosate formulation on migrating neutrophils in zebrafish” to “As far as we know, this is the first study to evaluate this new glyphosate formulation and its effect on neutrophil migration in zebrafish”. Indeed, the formulations maintain the main active component for decades but change its adjuvants which can result in different toxicities as referred in the reply to the point 3.

13) Please replace “Copper exposure” to “copper sulphate exposure” throughout the manuscript. Copper sulphate is a positive control for oxidative stress.

Response: As suggested by the reviewer, we modified the sentences in the text that contained only Copper and added copper sulfate.

14) Conclusion section is repetitive of result. It mostly talks about future studies needed. Having the answers for the raised issues would have been an advancement in the previous knowledge.

Response: As suggested by the reviewer, some excerpts of the conclusion were changed for better interpretation.

15) Please rephrase the sentence “it will allow for a ……” in line 329-330.

Response: As suggested by the reviewer, the sentence was rephrased.

16) Please revise manuscript thoroughly for typos and syntax errors.

Response: The text was revised by a native English person in order to correct errors and typos.

Round 2

Reviewer 2 Report

I have following points-

  1. Title still sounds conclusive and implies that RF exposure would cause neutrophil migration, which is not the case. So, it would be misleading. Please remove it from the title or rephrase the title to avoid conclusive message.  

  1. It is totally fine to letters or asterisks to indicate significance in multiple comparison set up as long as these are properly defined and used consistently. For figure 1 they have used a and b. For figure 2B, authors have used a, c, ab and bc in first panel (ROS) and then a, c, bc, ab and abc for second panel (SOD). Is there any reason to use double letter or triple letter code instead of all single letters a, b, c, d etc. Please define.

  1. Please cite the figure panel number along with the result description text in result section.
